# Low-Carbon Technology Innovation Decision Making of Manufacturing Companies in the Industrial Internet Platform Ecosystem

Hongxia Zhao, Guangming Xu, Lu Liu, Changchun Shi and Huijuan Zhao *

College of Economics and Management, Qingdao University of Science and Technology, Qingdao 266061, China
* Correspondence: zhj.smart@126.com

**Abstract:** Low carbon has become a highly relevant topic in today's society, particularly for manufacturing enterprises. To gain insight into how manufacturing enterprises embedded in the industrial internet platform make decisions regarding low-carbon technology innovation, this article examines the service quality of the platform, the low-carbon preferences of the manufacturing enterprises, and government subsidy factors. A platform ecological system game model, comprised of a single manufacturing enterprise and an industrial internet platform, is then established. The results indicate that, under the model's assumptions, the decarbonization of production can only occur when the cost of low-carbon innovation is below a specific threshold. Decentralized decision making is more effective in promoting low-carbon innovation by the manufacturing enterprises when the cost of low-carbon technology innovation is low. The greater the service quality of the industrial internet platform, the stronger the positive influence of the low-carbon preferences of users and government subsidies on the low-carbon innovation level of the manufacturing enterprises. This study offers useful decision-making advice for both the industrial internet platform and the manufacturing enterprises.

**Keywords:** low-carbon technology innovation; platform service quality; low-carbon preference; government subsidy



## 1. Introduction

The issue of carbon emissions has become increasingly pressing as the need for environmental protection becomes more urgent. It has garnered significant attention from countries around the world, as evidenced by the multitude of policies and plans that are being implemented to address this problem [1]. For instance, China has introduced its "double carbon" strategy, which aims to reach a "carbon peak" by 2030 and achieve carbon neutrality by 2060 [2]. The United States, having returned to the Paris Agreement, has committed to prioritizing clean energy and public transportation in the future, with a goal of achieving comprehensive carbon neutrality by 2050 [3]. Similarly, the European Union's "Fit for 55" plan aims to reduce carbon emissions by 55% by 2030 in order to combat climate change [4]. The impact of social responsibility and environmental policies has resulted in a wide range of research conducted by scholars in various fields, including carbon pricing and trading, green technology innovation, and international cooperation in emissions reduction [5–8]. As such, the question of how manufacturing firms can adapt to social development and policy systems in order to reduce their carbon emissions has become a topic of significant interest among scholars [9].

Manufacturing companies play a crucial role in addressing the issue of carbon emissions, as their production processes are inherently energy-intensive and often result in significant emissions. The increasing pressure from governments and society for environmental protection has resulted in a growing number of manufacturing companies investing in low-carbon technology innovation [10]. This refers to the promotion and application of new technologies, processes and products that effectively reduce carbon emissions.

These technologies include the adoption of new energy technologies, renewable energy sources, energy-saving measures, carbon capture and storage, as well as the enhancement of industrial energy efficiency and low-carbon product research and development. The implementation of these technologies can not only help to decrease carbon emissions, but also mitigate the impacts of climate change, support economic structural transformation, and promote sustainable development [11]. Generally, low-carbon technology innovation encompasses both technical innovation in production processes and the enhancement of product sustainability, which can ultimately lead to reduced carbon emissions, as well as improved production efficiency and reduced costs [12]. Manufacturing companies are embedded within complex systems, where a variety of factors can have a profound impact on decision making. Accordingly, the study of factors that influence the extent of low-carbon technology innovation within manufacturing companies has become a key area of research among scholars. Currently, it is widely acknowledged that government subsidies, environmental regulations, user low-carbon preferences, and the incorporation of third-party forces are crucial factors that can shape the degree of low-carbon technology innovation within manufacturing companies [13–15].

The industrial internet platform represents the convergence of traditional industry and information technology, enabling the integration of industrial equipment, production lines, production processes, logistics, transportation, sales, and other related areas through internet technology [16]. This results in capabilities such as real-time monitoring, data analysis, and automation control. The adoption of the industrial internet platform enables improved production efficiency, cost reduction, enhanced product quality, and increased customer satisfaction. Furthermore, it facilitates the sharing of manufacturing resources and capabilities, through effective order matching based on manufacturing companies' production behavior, which allows for adjustments in production efficiency, increases in market share and ultimately, better achievement of transformation and upgrading for manufacturing companies [17–19]. Research on industrial internet platforms primarily focuses on the following areas: (1) Industry 4.0's concept of smart manufacturing and the ability of industrial internet platforms to facilitate the realization of smart manufacturing systems, factory intelligence, and industrial big data; (2) real-time monitoring, data analysis, and remote control through the support of Internet of Things technology provided by industrial internet platforms; (3) the utilization of machine learning and artificial intelligence to analyze and predict through the vast amounts of data provided by industrial internet platforms, ultimately leading to intelligent scheduling and output planning.

The industrial internet platform holds a dominant position in the overall ecosystem, and the platform's policies and service quality directly impact the decision making of manufacturing companies in the ecosystem [20]. Scholars both domestically and internationally have focused their research on the effects of the industrial internet platform on manufacturing companies in areas such as production efficiency, product quality, production flexibility, and data analysis capabilities. For example, through the industrial internet platform, it is possible to monitor and control the production process, improve production efficiency, and reduce energy consumption [21]. The real-time monitoring of equipment and production status allows for the prompt identification and resolution of issues, leading to improved product quality [22]. The collection and analysis of a vast amount of production data using data analysis tools provides valuable information for decision-making support [23]. Therefore, it is widely agreed that the industrial internet platform can promote technological innovation in its ecosystem's manufacturing companies. However, research on the effects of the industrial internet platform on low-carbon technology innovation in manufacturing companies is limited.

This paper explores the decision-making process of manufacturing enterprises regarding low-carbon technology innovation, a crucial aspect in today's society where low-carbon initiatives are gaining prominence. Given that manufacturing enterprises contribute significantly to carbon emissions, it is important to understand how they approach low-carbon technology innovation. This research focuses on manufacturing enterprises embedded

in the industrial internet platform ecosystem and considers both government subsidies and user enterprises' low-carbon preferences, as well as the quality of industrial internet platform services. A game model is developed to analyze the impact of various policies in the platform ecosystem on low-carbon technology innovation by manufacturing enterprises. The paper also investigates the implications of independent decision making and centralized decision making for low-carbon technology innovation. The findings of this study offer valuable insights for the industrial internet platform and manufacturing enterprises as they navigate the challenges and opportunities of low-carbon innovation.

## 2. Literature Review

### 2.1. Low-Carbon Technology Innovation

Low-carbon technology innovation is an integral part of industrial energy conservation and emission reduction, and is a necessary step in the future development of manufacturing companies. Although perspectives on the definition of low-carbon technology innovation may vary, the significant role it plays in the growth and evolution of manufacturing companies has been widely acknowledged [24]. Manufacturing companies are the primary actors in technological innovation, and are also the driving force behind low-carbon technology innovation. Through the implementation of low-carbon technology, manufacturing companies can improve energy efficiency and reduce carbon emissions, thus positively impacting both the environment and their bottom line [25]. Adopting low-carbon technology not only reduces carbon emissions, but also makes products more appealing to consumers, increasing market share and promoting overall growth and development [26].

Scholars both domestically and internationally have conducted a significant amount of research on low-carbon technology innovation. The research mostly focuses on whether low-carbon technology innovation can promote the performance of manufacturing companies or analyzes the factors that influence manufacturing companies' adoption of low-carbon technology innovation. For example, Li et al. [27] conducted extensive research on low-carbon technology innovation and its effects on the performance of manufacturing companies. Their findings show that low-carbon technology innovation has a positive impact on performance, with green core competitiveness playing a mediating role and enterprise size playing a regulating role. Huang et al. [28] explored the relationship between technology coupling and low-carbon technology innovation, providing recommendations for manufacturing companies to choose appropriate methods and implement low-carbon technology innovation in the context of external pressures and internal acceptance. Li et al. [29] also studied the relationship between green finance and corporate low-carbon technology innovation, finding that green finance can significantly enhance low-carbon technology innovation and that corporate social responsibility plays a positive regulating role.

Manufacturing companies that rely solely on their own actions to undertake low-carbon technology innovation may face the risk of market failure [30,31]. Both reality and research indicate that guidance and support from the government is necessary. As a policy maker and enforcer, the government plays an important role in manufacturing companies' low-carbon technology innovation, through policies such as carbon trading, environmental regulation, and subsidies and incentives. Studies, such as those by Gao et al. [32], found that the carbon emission trading system affects enterprises' low-carbon technology innovation decisions, and identified five significant impact factors that are important to consider in the design of the carbon emission trading system. Chen et al. [33] also used empirical methods to prove that both environmental uncertainty and environmental regulation promote the low-carbon technology innovation of enterprises, with the latter highlighting the relationship between environmental uncertainty and low-carbon technology innovation. Additionally, research by Guan et al. [34] showed that different subsidy modes can impact low-carbon technology innovation, and that a flexible innovative subsidy combination should be adopted under different circumstances.

The decision-making mode of manufacturing enterprises has significant repercussions on various aspects of the enterprise, including product pricing, cost sharing, and low-carbon

decision-making research, which have garnered significant attention from researchers. For instance, Fuli et al. [35] investigated the impact of decision-making methods on carbon reduction through a two-stage supply-chain game model under centralized, decentralized, and coordinated decisions. Jian et al. [36] analyzed the decision making and motivations of supply-chain members in a green closed supply chain through a Stackelberg game model under centralized and decentralized decisions, and found that a well-designed profit-sharing contract can lead to sustainable economic and environmental outcomes. Liu et al. [37] studied the impact of supply-chain structure on the optimal game decision in a two-tier supply-chain system through a comparison of centralized and decentralized decisions in terms of optimal wholesale price, product carbon emissions, and retailer's optimal sales price. In light of these findings, this paper aims to examine the low-carbon technology innovation of manufacturing enterprises within the industrial internet platform ecology under different decision-making modes.

*2.2. The Service Quality of the Industrial Internet Platform*

The theories and technologies of industrial internet platforms are continuously evolving and developing, and have attracted widespread attention from scholars both domestically and internationally. Research on the impact of industrial internet platforms on manufacturing companies' technological innovation has mainly identified two main perspectives. On one hand, industrial internet platforms break down barriers and limitations in terms of enterprise boundaries and resources, creating an environment that supports the efficient sharing of product design resources and the integration of resources in the manufacturing process. This in turn shortens product development cycles and promotes technological innovation within the enterprise [38]. On the other hand, industrial internet platforms provide real-time data for managing resources, respond to diverse customer demands, and give rise to various new manufacturing models. This accelerates the development speed of manufacturing companies and drives technological innovation [39].

As the manufacturing industry shifts from mass production to personalized customization, the platform services of industrial internet platforms such as customer demand mining and supply–demand matching have garnered significant attention from scholars worldwide [40]. In traditional marketing research, service quality is defined as the difference between customers' perceived service and their expected service after experiencing actual service. For internet platforms, platform service quality is considered a key indicator of the platform operator's competitiveness, as improving service quality can attract more users to join the platform ecosystem [41]. Scholars such as Xue et al. [42] have explored the impact of platform service quality on the market through theoretical analysis and game models, highlighting the crucial role that platform service quality plays in determining platform pricing strategies. Other scholars, such as Liu et al. [43], have revealed through modeling that platform service quality can influence users' trust or distrust towards the platform, influencing perception and curiosity, with both dimensions of platform service quality having a positive impact on users' trust in the platform. Research on crowd delivery platforms by He et al. [44] also indicated that different service quality levels should be adopted at different stages of platform development, and that different service quality levels lead to different costs, which can affect the platform's pricing strategy. Overall, platform service quality has emerged as a critical element of industrial internet platforms and is attracting a lot of attention from researchers worldwide.

Currently, research on how industrial internet platforms promote low-carbon technological innovation in manufacturing companies is severely lacking, but in reality, the development of industrial internet platforms is rapid, and significant progress has been made in promoting low-carbon technological innovation in the manufacturing industry. For instance, the COSMOplat platform established an energy business sector—Smart Energy—which focuses on energy management and new energy technology research and development by utilizing advanced technologies such as big data and 5G. By utilizing cutting-edge techniques and matching appropriate resources, the platform creates

customized low-carbon solutions for companies, thereby promoting their low-carbon technological innovation and the construction of low-carbon production factories. Additionally, the COSMOplat platform provides scene marketing services for Haier Smart Home, significantly expanding the sales channels for low-carbon products, resulting in a significant increase in orders. This increased revenue, in turn, promotes the low-carbon technological innovation of the company [45].

Previous research on industrial internet platforms has primarily focused on the technology of the platform and its impact on corporate efficiency and decision making. Studies on platform service quality have mainly examined its impact on consumer behavior and pricing strategies. Despite the proven ability of these platforms to drive technological innovation within enterprises [46], there is limited research on their effect on low-carbon technology innovation. This paper seeks to fill this gap by considering the quality of industrial internet platform services and analyzing their impact on low-carbon technology innovation in manufacturing enterprises, while also examining the role of government subsidies and user preferences. Additionally, the paper explores optimal low-carbon technology innovation levels under different decision-making modes between the platform and manufacturing enterprise.

## 3. Modeling

This study investigates a system comprising an industrial internet platform and a manufacturing enterprise that is embedded within the ecosystem of the platform. The government provides subsidies to manufacturing enterprises to decrease carbon emissions during the production process. These manufacturing enterprises utilize low-carbon innovation to create low-carbon products, which are then delivered to user companies via the industrial internet platform. When a user company submits a demand on the platform, it is matched with a suitable manufacturing enterprise. During this process, the platform charges a commission from the manufacturing enterprise. The platform operator, being the leader of the ecosystem, decides the commission charged. Meanwhile, the manufacturing enterprise, as a follower of the ecosystem, determines the product's price and the degree of low-carbon innovation. The decision-making sequence of the system is as follows: the platform, with the objective of maximizing its own benefits, decides on the unit product commission $\rho$, constituting the first stage of the game. Subsequently, the manufacturing enterprise, with the objective of maximizing its own benefits, decides on the product's selling price $p$ and low-carbon innovation degree $r$, constituting the second stage of the game. The model's parameters are detailed in Table 1. The theoretical mechanism diagram of low-carbon technology innovation of manufacturing enterprises under the industrial internet platform is in Figure 1.

**Table 1.** Parameters and descriptions.

| Notation | Description |
| :---: | :---: |
| $a$ | Basic market capacity |
| $p$ | Product sales price |
| $C_z$ | Unit production cost |
| $q$ | Sales volume |
| $r$ | Low-carbon technology innovation level |
| $s$ | Platform service quality |
| $k$ | Low-carbon technology innovation cost coefficient |
| $h$ | Platform service cost coefficient |
| $\lambda$ | Price sensitivity coefficient |
| $\beta$ | Low-carbon preference coefficient |
| $\mu$ | Impact coefficient of platform service quality |
| $\rho$ | Commission ratio per unit product |
| $\theta$ | Government subsidy coefficient |
| $\pi_m, \pi_p$ | Benefits of manufacturing enterprises, Benefits of platform |

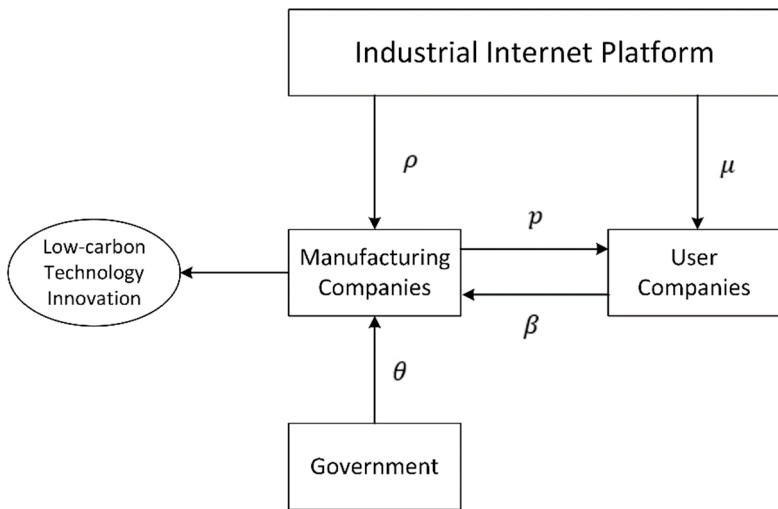

**Figure 1.** Theoretical mechanism diagram.

The basic assumptions of the model are as follows:

1. There is complete symmetry of market information between the manufacturing enterprise and the platform, and the low-carbon product output of the manufacturing enterprise can be completely cleared under the platform's matching, that is, the platform can always match and meet market demand;

2. Carbon emissions only occur in the production stage, and manufacturing enterprises reduce emissions by developing low-carbon production technology. The cost function for the low-carbon innovation of manufacturing enterprises is $C_r = \frac{1}{2}kr^2$;

3. The platform provides basic services for matching manufacturing enterprises to user companies, and the services provided by the platform are fully customized according to user company needs. In fact, in recent years, there have been several examples of industrial internet platforms, such as the COSMOplat platform, that provide professional and customized low-carbon transformation solutions; the platform's service cost function $C_s = \frac{1}{2}hs^2$;

4. User companies have a certain degree of low-carbon preferences, and demand is composed of price, low-carbon innovation degree, platform service quality, price sensitivity coefficient, low-carbon preference coefficient, and platform service quality coefficient, making the demand as shown in Formula (1);

5. It is assumed that the low-carbon preference coefficient of user companies satisfies: $\lambda\theta - 3\beta > 0$. According to the above assumptions, it can be obtained that the manufacturing enterprise's profit function and the platform's profit function are as shown in Formulas (2) and (3).

$$q = a - \lambda p + \beta r + \mu s \tag{1}$$

$$\pi_m = (p - C_z - \rho + r\theta)q - \frac{1}{2}kr^2 \tag{2}$$

$$\pi_p = \rho q - \frac{1}{2}hs^2 \tag{3}$$

By using the reverse solution method [47], first, Formula (1) is substituted into Formula (2) and its first derivative is taken with respect to the selling price $p$ to obtain the optimal selling price $p^*$; then $p^*$ is substituted into Formula (3) and its first derivative taken with respect to the commission ratio $\rho$ to obtain the optimal commission ratio $\rho^*$; finally, $p^*$ and $\rho^*$ are substituted into Formulas (1)–(3) to obtain the optimal market capacity $q^*$, the optimal manufacturing enterprise revenue $\pi_m^*$, and the optimal platform revenue $\pi_p^*$. The results are shown in Table 2.

**Table 2.** Solution result.

| Definition | Result |
|---|---|
| Market capacity | $q = \frac{a+\beta r+\mu s-\lambda C_z+\lambda r\theta}{4}$ |
| Selling price | $p = \frac{3a+3\beta r+3\mu s+\lambda C_z-\lambda r\theta}{4\lambda}$ |
| Commission rate | $\rho = \frac{a+\beta r+\mu s-\lambda C_z+\lambda r\theta}{2\lambda}$ |
| Benefits of manufacturing enterprises | $\pi_m = \frac{(a+\beta r+\mu s-\lambda C_z+\lambda r\theta)^2}{16\lambda} - \frac{1}{2}kr^2$ |
| Benefits of platform | $\pi_p = \frac{(a+\beta r+\mu s-\lambda C_z+\lambda r\theta)^2}{8\lambda} - \frac{1}{2}hs^2$ |

Under the premise that the parameters and variables are meaningful, that is $p > 0$, $q > 0$, $\rho > 0$, it can be known that the range of $r$ needs to meet the relationship Equation (4).

$$\frac{\lambda C_z - (a + \mu s)}{\beta + \lambda\theta} < r < \frac{3(a + \mu s) + \lambda c_z}{\lambda\theta - 3\beta} \tag{4}$$

In order to investigate the strategies that the platform and manufacturing enterprises can adopt to optimize the level of low-carbon innovation of products and to examine the impact of government subsidies, platform service quality, and user company low-carbon preferences on the level of low-carbon innovation, this study adopts the research framework of traditional supply chain and conducts an analysis of dispersed and centralized decision making [48,49]. Under dispersed decision making, both the industrial internet and manufacturing enterprises aim to maximize their own interests. Under centralized decision making, the two parties jointly aim to optimize the benefits of the platform ecosystem.

## 4. Decision Analysis

### 4.1. Dispersed Decision Making

In this decision scenario, the low-carbon innovation level *r* serves as an independent decision variable for the manufacturing firm, and the firm's profit function is $\pi_m = (a + \beta r + \mu s - \lambda c_z + \lambda r\theta)^2/16\lambda - 0.5kr^2$. According to the first-order optimization condition, by setting $\partial\pi_m/\partial r = 0$, the optimal low-carbon innovation level is obtained as shown in Equation (5).

$$r^* = \frac{(\beta + \lambda\theta)(a + \mu s - \lambda c_z)}{8\lambda k - (\beta + \lambda\theta)^2} \tag{5}$$

It is easy to prove that when $k < \frac{(\beta+\lambda\theta)^2}{8\lambda}$, $\pi_m$ takes on a minimum value at $r^*$, and when $k > \frac{(\beta+\lambda\theta)^2}{8\lambda}$, $\pi_m$ takes on a maximum value at $r^*$.

According to the range of values of *r* in Equation (4) and considering that $r \leq 1$, by analyzing the conditions required for *r* to take on its maximum value under different values of *a* and *k*, it can be concluded that *r* has four stationary values: $r_1 = [\lambda c_z - (a + \mu s)]/(\beta + \lambda\theta)$; $r_2 = [3(a + \mu s) + \lambda c_z]/(\lambda\theta - 3\beta)$; $r_3 = 1$; $r_4 = [(\beta + \lambda\theta)(a + \mu s - \lambda c_z)]/[8\lambda k - (\beta + \lambda\theta)^2]$. Among them, due to $r = r_1$ and $r = r_2$, the values of *p*, *q*, and $\rho$ are 0, so these two possibilities are discarded.

It is easy to prove that when either of the two conditions shown in Equation (6) is met, $\pi_m$ reaches its maximum value at $r = 1$; when either of the two conditions shown in Equation (7) is met, $\pi_m$ reaches its maximum value at $r = r_4$, where the values of $K_1$, $K_2$, and $K_3$ are as follows.

$$\begin{cases} K_1 = \frac{(a+\beta+\mu s-\lambda c_z+\lambda\theta)(\beta+\lambda\theta)^2}{8\lambda[(\beta+\lambda\theta)-(a+\mu s-\lambda c_z)]} \\ K_2 = \frac{(\beta+\lambda\theta)(a+\mu s-\lambda C_z)+(\beta+\lambda\theta)^2}{8\lambda} \\ K_3 = \frac{(\beta+\lambda\theta)\{(\lambda\theta-3\beta)(a+\mu s-\lambda c_z)+[3(a+\mu s)+\lambda c_z](\beta+\lambda\theta)\}}{8\lambda[3(a+\mu s)+\lambda c_z]} \end{cases}$$

Further analysis of Equation (6) shows that when either of the conditions in Equation (8) is met, $\pi_m$ reaches its maximum value at $r = 1$, and the corresponding product demand, sales price, and platform commission ratio are shown in Table 3. When either of the conditions in Equation (9) is met, $\pi_m$ reaches its maximum value at $r = \frac{(\beta+\lambda\theta)(a+\mu s-\lambda c_z)}{8\lambda k-(\beta+\lambda\theta)^2}$, and the corresponding product demand, sales price, and platform commission ratio are shown in Table 4.

$$\begin{cases} \max\left[\frac{\lambda\theta-3\beta-3\mu s-\lambda c_z}{3}, \lambda c_z - \mu s - \beta - \lambda\theta\right] < a < \lambda c_z - \mu s & and \ k < K_1 \\ a \geq \lambda c_z - \mu s & and \ k < K_2 \end{cases} \quad (6)$$

$$\begin{cases} \max\left[\lambda c_z - \mu s, \frac{\lambda\theta-3\beta-3\mu s-\lambda c_z}{3}\right] < a & and \ k > K_2 \\ \lambda c_z - \mu s < a < \frac{\lambda\theta-3\beta-3\mu s-\lambda c_z}{3} & and \ k > K_3 \end{cases} \quad (7)$$

$$\begin{cases} \lambda C_z - \mu s - \beta - \lambda\theta < a < \lambda C_z - \mu s \ and \ k < K_1 & if \ \theta < \frac{3\beta+4\lambda C_z}{\lambda} \\ a \geq \lambda C_z - \mu s \ and \ k < K_2 & if \ \theta < \frac{3\beta+4\lambda C_z}{\lambda} \\ a > \frac{\lambda\theta-3\beta-3\mu s-\lambda C_z}{3} \ and \ k < K_2 & if \ \theta > \frac{3\beta+4\lambda C_z}{\lambda} \end{cases} \quad (8)$$

$$\begin{cases} a \geq \lambda C_z - \mu s \ and \ k > K_2 & if \ \theta < \frac{3\beta+4\lambda C_z}{\lambda} \\ a > \frac{\lambda\theta-3\beta-3\mu s-\lambda C_z}{3} \ and \ k > K_2 & if \ \theta > \frac{3\beta+4\lambda C_z}{\lambda} \\ \lambda C_z - \mu s \leq a < \frac{\lambda\theta-3\beta-3\mu s-\lambda C_z}{3} \ and \ k > K_3 & if \ \theta > \frac{3\beta+4\lambda C_z}{\lambda} \end{cases} \quad (9)$$

**Table 3.** Solution result ($r = r_3$).

| Definition | Result |
|---|---|
| Market capacity | $q = \frac{a+\beta r+\mu s-\lambda C_z+\lambda r\theta}{4}$ |
| Selling price | $p = \frac{3a+3\beta r+3\mu s+\lambda C_z-\lambda r\theta}{4\lambda}$ |
| Commission rate | $\rho = \frac{a+\beta r+\mu s-\lambda C_z+\lambda r\theta}{2\lambda}$ |
| Benefits of manufacturing enterprises | $\pi_m = \frac{(a+\beta r+\mu s-\lambda C_z+\lambda r\theta)^2}{16\lambda} - \frac{1}{2}kr^2$ |
| Benefits of platform | $\pi_p = \frac{(a+\beta r+\mu s-\lambda C_z+\lambda r\theta)^2}{8\lambda} - \frac{1}{2}hs^2$ |

**Table 4.** Solution result ($r = r_4$).

| Definition | Result |
|---|---|
| Market capacity | $q = \frac{2\lambda k(a-\lambda C_z+\mu s)}{8\lambda k-(\beta+\lambda\theta)^2}$ |
| Selling price | $p = \frac{(6k-\lambda\theta^2-\beta\theta)(a+s\mu)+C_z(2k\lambda-\beta^2-\beta\theta\lambda)}{8\lambda k-(\beta+\lambda\theta)^2}$ |
| Commission rate | $\rho = \frac{4k(a-\lambda C_z+\mu s)}{8\lambda k-(\beta+\lambda\theta)^2}$ |
| Benefits of manufacturing enterprises | $\pi_m = \frac{k(a+\mu s-\lambda C_z)^2}{2[8\lambda k-(\beta+\lambda\theta)^2]}$ |
| Benefits of platform | $\pi_p = \frac{8\lambda k^2(a+\mu s-\lambda C_z)^2}{[8\lambda k-(\beta+\lambda\theta)^2]^2} - \frac{1}{2}hs^2$ |

Based on the above analysis, the following two propositions are established.

**Proposition 1.** *When $\theta < (3\beta + 4\lambda C_z)/\lambda$, ① $a \geq \lambda C_z - \mu s$, $k > K_2$; or when $\lambda C_z - \mu s \leq a < (\lambda\theta - 3\beta - 3\mu s - \lambda C_z)/3$, ① $k > K$ ② $a > (\lambda\theta - 3\beta - 3\mu s - \lambda C_z)/3$, $k > K_2$, under the dispersed decision-making scenario of the platform ecosystem, the optimal low-carbon innovation level of the manufacturing enterprise is $r = [(\beta + \lambda\theta)(a + \mu s - \lambda C_z)]/[8\lambda k - (\beta + \lambda\theta)^2]$.*

**Proposition 2.** *When $\theta < (3\beta + 4\lambda C_z)/\lambda$, ① $\lambda C_z - \mu s - \beta - \lambda\theta < a < \lambda C_z - \mu s$, $k < K_1$ ② $a \geq \lambda C_z - \mu s$, $k < K_2$; or when $\theta > (3\beta + 4\lambda C_z)/\lambda$, ① $a \geq \lambda C_z - \mu s$, $k < K_2$, the dispersed*

*decision making of the platform ecosystem can promote the manufacturing enterprises to achieve decarbonization production.*

**Conclusion 1.** *In a dispersed decision-making system within the platform ecosystem, the achievement of decarbonization by manufacturing companies remains contingent on the cost of low-carbon technology falling below a certain threshold, regardless of the intensity of government subsidies or the size of the market. This conclusion aligns with the reality that, in order to effectively incentivize manufacturing companies to adopt decarbonization practices, the cost of low-carbon innovation must be markedly low. Otherwise, manufacturing companies will tend to opt for moderate low-carbon innovation. Furthermore, government subsidies can play a role in reducing the cost of low-carbon R & D for manufacturing companies and encouraging low-carbon innovation [50]. However, attaining the goal of decarbonization will likely prove difficult without significant subsidies, as reliance on the commonly employed short-term, specific subsidies alone is inadequate.*

### 4.2. Centralized Decision Making

In centralized decision making, the industrial internet platform and manufacturing companies aim to maximize the system benefits, and the objective function of the platform ecosystem is as shown in Equation (10).

$$\pi = (p - C_z + r\theta)q - \frac{1}{2}kr^2 - \frac{1}{2}hs^2 \tag{10}$$

Similar to dispersed decision making, it can be known that the low-carbon technology innovation level of the product can reach its maximum value or even the decarbonization of production, and it is required to meet the following conditions.

When either of the conditions in Equation (11) is met, the optimal low-carbon innovation level is $r = 1$, and the corresponding product demand, sales price, and platform commission ratio are the same as in Table 3. When either of the conditions in Equation (12) is met, the optimal low-carbon innovation level is $r = r_5 = [3(\beta + \lambda\theta)(a + \mu s - \lambda C_z)]/[8\lambda k - 3(\beta + \lambda\theta)^2]$, and the corresponding product demand, sales price, and platform commission ratio are shown in Table 5.

$$\begin{cases} C_z - \mu s - \beta - \lambda\theta < a < \lambda C_z - \mu s \text{ and } k < K_1 & \text{if } \theta < \frac{3\beta + 4\lambda C_z}{\lambda} \\ a \geq \lambda C_z - \mu s \text{ and } k < K_2 & \text{if } \theta < \frac{3\beta + 4\lambda C_z}{\lambda} \\ a > \frac{\lambda\theta - 3\beta - 3\mu s - \lambda C_z}{3} \text{ and } k < K_2 & \text{if } \theta > \frac{3\beta + 4\lambda C_z}{\lambda} \end{cases} \tag{11}$$

$$\begin{cases} a \geq \lambda C_z - \mu s \text{ and } k > K_2 & \text{if } \theta < \frac{3\beta + 4\lambda C_z}{\lambda} \\ a > \frac{\lambda\theta - 3\beta - 3\mu s - \lambda C_z}{3} \text{ and } k > K_2 & \text{if } \theta > \frac{3\beta + 4\lambda C_z}{\lambda} \\ \lambda C_z - \mu s \leq a < \frac{\lambda\theta - 3\beta - 3\mu s - \lambda C_z}{3} \text{ and } k > K_3 & \text{if } \theta > \frac{3\beta + 4\lambda C_z}{\lambda} \end{cases} \tag{12}$$

**Table 5.** Solution result ($r = r_5$).

| Definition | Result |
| --- | --- |
| Market capacity | $q = \frac{2\lambda k(a + \mu s - \lambda C_z)}{8\lambda k - 3(\beta + \lambda\theta)^2}$ |
| Selling price | $p = \frac{(6k - 3\lambda\theta^2 - 3\beta\theta)(a + s\mu) + C_z(2k\lambda - 3\beta^2 - 3\beta\theta\lambda)}{8\lambda k - 3(\beta + \lambda\theta)^2}$ |
| Commission rate | $\rho = \frac{4k(a + \mu s - \lambda C_z)}{8\lambda k - 3(\beta + \lambda\theta)^2}$ |
| Benefits of manufacturing enterprises | $\pi_m = \frac{k(a + \mu s - \lambda C_z)^2[8\lambda k - 9(\beta + \lambda\theta)^2]}{[8\lambda k - 3(\beta + \lambda\theta)^2]^2}$ |
| Benefits of platform | $\pi_p = \frac{(a + \mu s - \lambda C_z)^2\{3(\beta + \lambda\theta)^2 + [8\lambda k - 3(\beta + \lambda\theta)^2]^2\}}{8[8\lambda k - 3(\beta + \lambda\theta)^2]^2} - \frac{1}{2}hs^2$ |

Based on the above analysis, the following two propositions are established.

**Proposition 3.** *When $\theta < (3\beta + 4\lambda C_z)/\lambda$, ① $a > \lambda C_z - \mu s$, $k > K_2$; or when $\theta > (3\beta + 4\lambda C_z)/\lambda$, ① $\lambda C_z - \mu s \leq a < (\lambda\theta - 3\beta - 3\mu s - \lambda C_z)/3$, $k > K_3$ ② $a > (\lambda\theta - 3\beta - 3\mu s - \lambda C_z)/3$, $k > K_2$, in the centralized decision making of manufacturing enterprises, the optimal low-carbon innovation level can be achieved with $r = [3(\beta + \lambda\theta)(a + \mu s - \lambda C_z)]/[8\lambda k - 3(\beta + \lambda\theta)^2]$.*

**Proposition 4.** *When $\theta < (3\beta + 4\lambda C_z)/\lambda$, ① $\lambda C_z - \mu s - \beta - \lambda\theta < a < \lambda C_z - \mu s$, $k < K_1$ ② $a \geq \lambda C_z - \mu s$, $k < K_2$; or when $\theta > (3\beta + 4\lambda C_z)/\lambda$, ① $a > (\lambda\theta - 3\beta - 3\mu s - \lambda C_z)/3$, $k < K_2$, in the centralized decision making of manufacturing enterprises, it can be achieved decarbonization production ( $r = 1$).*

**Conclusion 2.** *In a centralized decision-making system within the platform ecosystem, the achievement of decarbonization by manufacturing companies remains contingent on the cost of low-carbon innovation falling below a certain threshold, regardless of the intensity of government subsidies or the size of the market. This aligns with the dispersed decision-making scenario as the internalization of commission charges by the platform, while reducing costs within the ecosystem, still falls significantly short of the cost of low-carbon innovation, which remains the key determining factor. Therefore, it can be inferred that the cost of low-carbon technology must be relatively low for manufacturing companies to achieve decarbonization. Despite this, centralized decision making can still play a role in reducing costs within the ecosystem, allowing for the possibility of achieving decarbonization at relatively higher innovation costs in comparison to the dispersed decision-making scenario [51]. This highlights the potential for centralized decision making to promote decarbonization to a certain extent.*

*4.3. Comparative Analysis of Two Decision-Making Schemes*

**Proposition 5.** *When $\theta < (3\beta + 4\lambda C_z)/\lambda$, $a > \lambda C_z - \mu s$, ① $K_2 < k < 3K_2$, dispersed decision making will achieve a higher level of low-carbon innovation; ② $k > K_2$, centralized decision making will achieve a higher level of low-carbon innovation. When $\theta > (3\beta + 4\lambda C_z)/\lambda$, centralized decision making will achieve a higher level of low-carbon innovation.*

**Conclusion 3.** *In the case of limited government subsidies, dispersed decision making is more conducive to increasing the level of low-carbon innovation among manufacturing companies when the market capacity is large and the difficulty of low-carbon technology innovation is low. On the other hand, when market capacity is large but the difficulty of low-carbon technology innovation is high, centralized decision making is more beneficial in improving the level of low-carbon technology innovation among manufacturing companies.*

**Conclusion 4.** *In the case of significant government subsidies, centralized decision making is more conducive to increasing the level of low-carbon technology innovation among manufacturing companies. Firstly, low-carbon innovation necessitates a substantial investment in terms of cost, and under conditions of limited government subsidies, manufacturing companies can only justify such investments and attain desirable returns when market capacity is substantial. In scenarios where market capacity is large and the cost of low-carbon innovation for manufacturing companies is relatively low, competition among companies for low-carbon innovation is heightened, resulting in a higher level of low-carbon innovation among manufacturing companies under dispersed decision making. Conversely, when market capacity is large but the cost of low-carbon innovation is also high, the difficulty and risk associated with independent low-carbon innovation for companies is substantial, and centralized decision making proves to be more advantageous for manufacturing companies' low-carbon innovation efforts. Furthermore, when government subsidies are abundant, the risk associated with low-carbon innovation for manufacturing companies is significantly reduced, and market capacity becomes a less significant consideration. Centralized decision making can also aid in the sharing of costs among manufacturing companies, thus enabling a higher level of low-carbon innovation.*

In the dispersed decision-making scenario, by analyzing the dependency of $r$ on the related parameters, first, it is easy to calculate $\frac{\partial r}{\partial \beta} = \frac{(a+\mu s-\lambda C_z)[8\lambda k+(\beta+\lambda\theta)^2]}{[8\lambda k-(\beta+\lambda\theta)^2]^2} > 0$, $\frac{\partial^2 r}{\partial \beta \partial s} = \frac{\mu[8\lambda k+(\beta+\lambda\theta)^2]}{[8\lambda k-(\beta+\lambda\theta)^2]^2} > 0$. Therefore, conclusion 5 is established.

**Conclusion 5.** *The greater the low-carbon preference of user companies, the higher the level of low-carbon innovation among manufacturing companies. Additionally, the higher the quality of platform services, the greater the positive impact of user companies' low-carbon preference on the level of low-carbon innovation among manufacturing companies. This is because as the low-carbon preference of user companies increases, manufacturing companies will inevitably strive to meet user demands and gain a competitive edge by implementing higher levels of low-carbon technology innovation.*

Additionally, user companies with a strong low-carbon preference may lack understanding of low-carbon product information and struggle to find suitable low-carbon product suppliers. In this case, the involvement of the industrial internet is beneficial in improving these two situations [52]. When platform service quality is high, including pre-sales promotion and consultation of low-carbon products and post-sales guidance on usage, these services will make user companies more willing to connect with manufacturing companies on the platform [53], leading to a significant increase in the demand for low-carbon products within the platform ecosystem. Therefore, manufacturing companies will carry out more significant low-carbon innovation to meet this demand. Thus, the higher the quality of platform services, the greater the positive impact of user companies' low-carbon preference on the level of low-carbon innovation among manufacturing companies.

Secondly, by analyzing the dependency of $r$ on $\theta$ and $s$, it can be known that: $\frac{\partial r}{\partial \theta} = \frac{\lambda(a+\mu s-\lambda C_z)}{8\lambda k-(\beta+\lambda\theta)^2} + \frac{2\lambda(a+\mu s-\lambda C_z)(\beta+\lambda\theta)^2}{[8\lambda k-(\beta+\lambda\theta)^2]^2} > 0$, $\frac{\partial^2 r}{\partial \theta \partial s} = \frac{\lambda u}{8\lambda k-(\beta+\lambda\theta)^2} + \frac{2\lambda u(\beta+\lambda\theta)^2}{[8\lambda k-(\beta+\lambda\theta)^2]^2} > 0$. Therefore, conclusion 6 is established.

**Conclusion 6.** *The more government subsidies are provided, the higher the level of low-carbon innovation for manufacturing companies. Furthermore, the higher the quality of platform services, the greater the positive impact of government subsidies on the level of low-carbon innovation for manufacturing companies. Firstly, this is because when the government adopts subsidy policies for low-carbon innovation, manufacturing companies will carry out higher levels of low-carbon innovation in order to receive more subsidies. Secondly, when the quality of the platform's services is high, the sales of manufacturing companies' products will increase relatively, low-carbon products will be relatively more popular, making low-carbon innovation more profitable for manufacturing companies, and they will be more willing to engage in low-carbon innovation subjectively. Therefore, the government subsidy policy is more effective at this time.*

## 5. Analysis of Results

The analysis results of the model are verified through numerical simulation. Firstly, the relationship between low-carbon innovation cost and low-carbon innovation level is analyzed under dispersed and centralized decision-making scenarios. The parameters $\beta = 0.6$, $\lambda = 0.6$, $\theta = 0.5$, $\mu = 0.5$, $s = 5$, $C_z = 10$ are the same, and for the five parameter settings of $a = 5$ (—), $a = 8$ (-+-), $a = 11$ (—), $a = 14$ (-.-), $a = 17$ ( … ), the relationship between the low-carbon innovation level and low-carbon innovation cost of manufacturing companies is shown in Figure 2; for the five parameter settings of $a = 5$ (—), $a = 6$ (-+-), $a = 7$ (—), $a = 8$ (-.-), $a = 9$ ( … ), the relationship between the low-carbon innovation level and low-carbon innovation cost of manufacturing companies is shown in Figure 3.

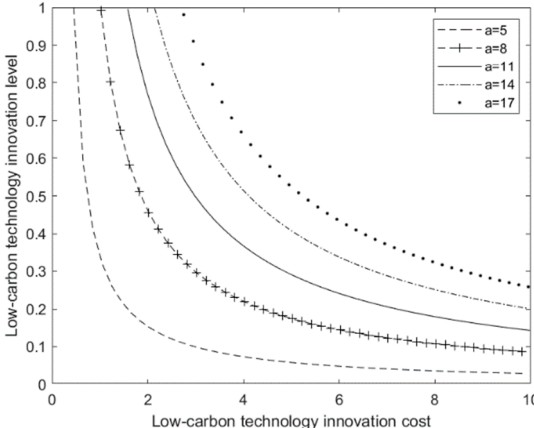

**Figure 2.** The relationship between low-carbon technology innovation cost and low-carbon technology innovation level in dispersed decision making.

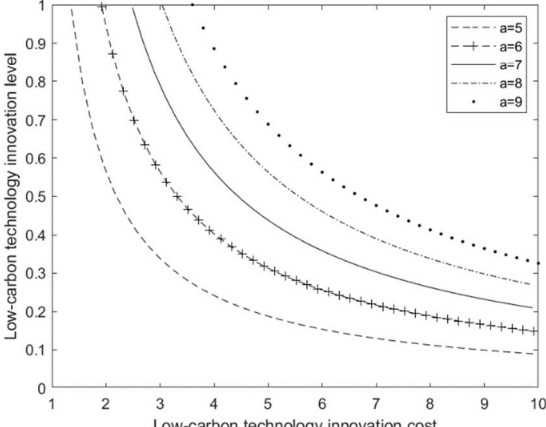

**Figure 3.** The relationship between low-carbon technology innovation cost and low-carbon technology innovation level in centralized decision making.

As can be seen from Figures 2 and 3, the level of low-carbon technology innovation decreases as the cost of low-carbon technology innovation increases in both decision-making scenarios. Furthermore, when the cost of low-carbon technology innovation is the same, the greater the market capacity, and the higher the level of low-carbon technology innovation. Regardless of whether the market capacity is large or small, only when the cost of low-carbon technology is low can manufacturing companies achieve decarbonization. In comparison, centralized decision making is more likely to achieve decarbonization, as it can achieve decarbonization at a relatively higher innovation cost.

Secondly, the analysis examined the relationship between user enterprises' low-carbon preferences and the level of low-carbon innovation under different platform service quality scenarios, wherein the parameters $k = 3$, $\lambda = 0.6$, $\theta = 0.5$, $\mu = 0.5$, $a = 5$, $C_z = 10$ are the same, and for the five sets of parameters $s = 5$ (—), $s = 8$ (-+-), $s = 11$ (—), $s = 14$ (-.-), $s = 17$ ( … ), the relationship between the low-carbon innovation level of manufacturing companies and user enterprises' low-carbon preferences is shown in Figure 4.

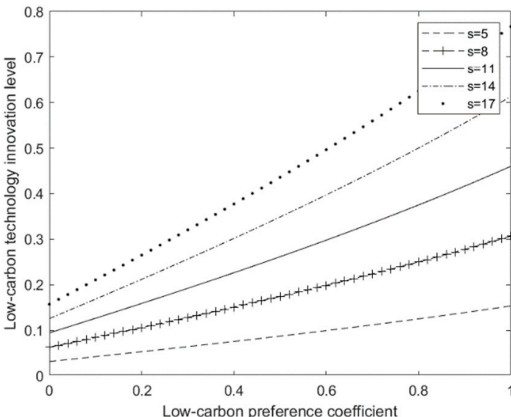

**Figure 4.** The relationship between low-carbon preference coefficient and low-carbon technology innovation level in centralized decision making.

As can be seen from Figure 4, the level of low-carbon innovation among manufacturing companies increases with the increase in low-carbon preference among user companies. Furthermore, at the same level of low-carbon preference, the higher the quality of platform service, and the higher the level of low-carbon innovation among manufacturing companies. Additionally, it can be seen that the higher the quality of platform service, the greater the slope of the curve, that is, the higher the quality of platform service, the greater the positive impact of user companies' low-carbon preferences on the low-carbon innovation of manufacturing companies.

Lastly, the relationship between the government subsidy coefficient and the level of low-carbon innovation among manufacturing companies was analyzed under different platform service quality scenarios. In this analysis, the parameters $k = 3, \beta = 0.6$, $\lambda = 0.6, \mu = 0.5, a = 5, C_z = 10$ are all the same, and for the five parameter settings $s = 5$ (—), $s = 8$ (-+-), $s = 11$ (—), $s = 14$ (-.-), $s = 17$ ( . . . ), the relationship between the level of low-carbon innovation among manufacturing companies and the government subsidy coefficient is shown in Figure 5.

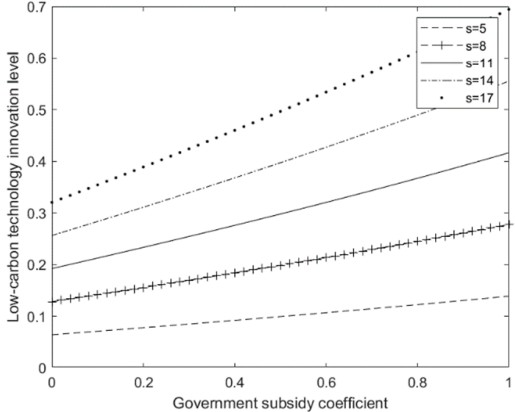

**Figure 5.** The relationship between government subsidy coefficient and low-carbon technology innovation level in centralized decision making.

As can be seen from Figure 5, the level of low-carbon innovation of manufacturing companies increases with the increase in government subsidy coefficients; given the same level of government subsidy coefficients, the higher the quality of service of the platform, the higher the level of low-carbon innovation of manufacturing companies. Additionally, it can be seen that when the quality of platform service is higher, the slope of the curve is greater, which means that the higher the quality of platform service, the greater the positive impact of government subsidy coefficients on the low-carbon innovation of manufacturing companies.



## 6. Conclusions

In recent years, the manufacturing sector has seen a growing trend towards low-carbon innovation as a means of enhancing the competitiveness of products. This top journal paper in Management presents a game model that examines the strategic interactions between two key decision-making entities: manufacturing enterprises and industrial internet platforms. The model takes into account factors such as user enterprises' low-carbon preferences, government subsidies, and platform service quality. Through rigorous analysis, this paper yields valuable insights and conclusions on the subject matter.

(1)  This study finds that for any given level of government subsidies and market demand, manufacturing companies will only be able to achieve decarbonization of production when the cost of low-carbon innovation is extremely low. This conclusion holds true in both centralized and dispersed decision-making scenarios. This highlights the challenging nature of achieving decarbonization of production for manufacturing companies with the current technology. Furthermore, as low-carbon technology advances and the cost of innovation decreases, it is expected that the level of low-carbon innovation among manufacturing companies will naturally increase. Thus, this study emphasizes the importance of investing in and promoting the advancement of low-carbon technology [54].

(2)  This study examines the impact of government low-carbon subsidies on the level of low-carbon innovation achieved by manufacturing companies. We find that under conditions of minimal government subsidies, optimal low-carbon innovation can only be achieved if the market demand for low-carbon products is substantial. In these circumstances, dispersed decision making is more effective in promoting low-carbon innovation when the cost of innovation is low, whereas centralized decision making is more effective when the cost of innovation is high. Conversely, under conditions of substantial government subsidies, centralized decision making is consistently more effective in promoting low-carbon innovation, regardless of the cost of innovation. In summary, this study highlights the multifaceted nature of achieving high levels of low-carbon innovation for manufacturing companies, and the importance of efforts from the government, platforms, and other sectors of society in meeting internal and external conditions such as the enhancement of low-carbon preferences among user enterprises and terminal users, an increase in government subsidies, and the empowerment and support of industrial internet platforms.

(3)  This study finds that a greater low-carbon preference among user enterprises leads to a higher level of low-carbon innovation among manufacturing companies. Additionally, we find that the higher the quality of service of the industrial internet platform, the greater the positive impact of user enterprises' low-carbon preferences on the low-carbon innovation of manufacturing companies. This is due to the fact that a higher-quality platform service leads to increased sales of low-carbon products for manufacturing companies, reducing the risk associated with low-carbon innovation and resulting in increased profits. Given the critical role that market acceptance plays in the low-carbon innovation of manufacturing companies [55], this study suggests that industrial internet platforms should work to increase the transparency of low-carbon products, promote the social benefits of low-carbon products, and enhance trust and acceptance among user enterprises.

(4)  The greater the government's subsidies for low-carbon innovation, the higher the level of low-carbon innovation among manufacturing companies. The quality of service provided by industrial internet platforms also plays a role in amplifying the impact of government subsidies on the low-carbon innovation of manufacturing companies. In practice, some companies have a strong inclination towards low-carbon innovation and government subsidies play a significant role in promoting their low-carbon technology innovation. However, some companies may engage in low-carbon innovation primarily to take advantage of government subsidies [56]. The introduction of industrial internet platforms can increase the popularity of low-carbon



innovative products, thereby increasing the subjective willingness of manufacturing companies to innovate and reducing the number of cases where companies are only motivated by government subsidies.

This research can bring the following management implications for the industrial internet platform and the manufacturing enterprises embedded within it:

(1)   Manufacturing enterprises should embrace low-carbon technology innovation when the cost is low, as the trend towards green and low-carbon products drives consumer demand. By prioritizing low-carbon technology innovation, companies can ensure long-term viability and competitiveness, without exposing themselves to financial risk. In this era of environmental awareness, it is crucial for enterprises to proactively adopt low-carbon technologies and pursue de-carbonization, even when costs are low.

(2)   For manufacturing enterprises, the pursuit of low-carbon technology innovation should be prioritized when government subsidies are strong or consumer preferences for low-carbon products are high. The availability of government subsidies can provide a cushion for manufacturing enterprises to invest in low-carbon technology innovation without having to worry excessively about the cost, thus avoiding financial stress. On the other hand, high consumer preferences for low-carbon products can drive enterprises to innovate in this area, lest they risk losing market share and hindering their development.

(3)   In order to enhance low-carbon technology innovation within the industrial internet platform ecosystem, it is crucial to improve the quality of the platform's service. A higher level of service quality will not only expand the impact of government subsidies and consumer preferences for low-carbon, but also facilitate the promotion of low-carbon technology innovation among manufacturing enterprises. Furthermore, a high standard of service quality will increase consumer recognition of manufacturing enterprises and their low-carbon products within the platform ecosystem.

(4)   This study has several limitations. This paper considers manufacturing companies and industrial internet platforms as decision-making subjects and only examines dispersed and centralized decision-making scenarios. In the future, other scenarios should be considered, such as the altruistic behavior or fairness concerns of decision-making subjects, and a comparison of the optimal levels of low-carbon technology innovation and implementation conditions in other contexts. Additionally, future research could include the government as a decision-making subject to explore the optimal subsidy policy for promoting social welfare. Furthermore, future research could also include the quality of platform service as a decision variable to analyze the strategic decision making of industrial internet platform companies in the context of low-carbon innovation.

**Author Contributions:** Conceptualization, H.Z. (Hongxia Zhao), G.X., L.L., C.S. and H.Z. (Huijuan Zhao); methodology, H.Z. (Huijuan Zhao); software, G.X.; validation, H.Z. (Hongxia Zhao); formal analysis, H.Z. (Hongxia Zhao); investigation, G.X.; resources, H.Z. (Hongxia Zhao); data curation, G.X.; writing—original draft preparation, H.Z. (Hongxia Zhao), G.X. and L.L.; writing—review and editing, C.S.; visualization, H.Z. (Hongxia Zhao); supervision, H.Z. (Huijuan Zhao); project administration, H.Z. (Huijuan Zhao). All authors have read and agreed to the published version of the manuscript.

**Funding:** The General programs of the National Social Science Foundation of China: 17BGL027.

**Institutional Review Board Statement:** Not applicable.

**Informed Consent Statement:** Not applicable.

**Data Availability Statement:** Not applicable.

**Acknowledgments:** Special thanks to those who participated in the writing of this paper.

**Conflicts of Interest:** The authors declare no conflict of interest.

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
