# Peer review of "Low-Carbon Technology Innovation Decision Making of Manufacturing Companies in the Industrial Internet Platform Ecosystem"

_sustainability, doi:10.3390/su15043555_

Round 1

Reviewer 1 Report

In order to study how manufacturing enterprises embed low carbon technology innovation decisions in the form of industrial Internet platforms, this paper considers the service quality of industrial Internet platforms, low carbon preferences of manufacturing enterprises, government subsidies and other factors, and establishes a platform ecosystem game model for a single manufacturing enterprise and industrial Internet platforms. The writing of this paper is standardized, the content is substantial, the mathematical model is used properly, and the research of this paper is reliable. Specific comments are as follows

1.In the introduction, it is suggested to further emphasize the general importance and necessity of this study, and concisely explain the innovation points of this paper.

2.The literature review part is lack of sorting out the enterprise decision-making mode, and it is recommended to add.

3.This paper has carried out a lot of mathematical derivation, but it lacks the overall mechanism of low-carbon technology innovation of enterprises under the industrial Internet platform. It is suggested to supplement the theoretical mechanism diagram to show it more clearly

4.It is suggested to put forward more targeted policy recommendations based on the research results of the paper to improve the practical value of the paper

Reviewer 2 Report

1.The introduction is poor. It is suggested to clearly introduce the research ideas, research methods, research issues, and purpose of the paper.

2. In model,many decision parameters of the industrial Internet platform are considered in this paper. However, I suggest that decision parameters should Be explained with realistic meaning.

3. The role of the industrial internet platform mentioned in the article is similar to that of e-commerce platform, what is the difference between the industrial internet platform and e-commerce platform? It is suggested to clearly distinguish the relationships.

4. In the manuscript, the simulation parameters are set too arbitrarily. The author should explain the source of the simulation parameters.

5. The conclusion section is poorly written, and the authors do not discuss the differences between the findings and the existing results, nor do they state the main contribution of the manuscript.

6. The literature review is recommended to be reorganized, the current literature does not describe the source and innovation of this study.

Reviewer 3 Report

This article considers factors such as the quality of service of industrial internet platforms, the low-carbon preferences of manufacturing enterprises, and government subsidies, and establishes a platform ecosystem game model consisting of a single manufacturing enterprise and an industrial internet platform. The article has an interesting approach, however there are some issues that authors must correct:

1. The objective of the article is not clear either in the abstract or in the introduction.

2. The abstract does not bring a contextualization of the topic to be addressed, nor does it highlight the importance of developing this article.

3. In the introduction there are some statements with numerical data that do not have the reference of origin of such data.

4. The authors do not make it clear in the introduction what practical contributions are expected from this article.

5. How did the authors select the parameters shown in Table 1? It is necessary to detail this and show the origin. Are there other indicators besides these? How to justify these choices?

6. The authors say that the study investigated an industrial internet platform and a manufacturing enterprise that is embedded within the ecosystem of the platform. It is necessary to present more details about this, where the company is located, some idea of the size of the platform and the company, which area the company operates, among other details.

7. Does this mathematical detail presented to solve the modeling originate from any method? More details need to be provided on how this was chosen and why.

8. Sections 4, 5 and 6 are well organized and well written, highlighting the relevance and implications of this research.

Round 2

Reviewer 3 Report

I congratulate the authors for the good review work done. The authors were concerned with meeting the suggestions and answered the reviewer's questions satisfactorily. The article can be accepted for publication.